# Design and Development of a Hemorrhagic Trauma Simulator for Lower Limbs: A Pilot Study

**DOI:** 10.3390/s21113816

**Published:** 2021-05-31

**Authors:** Blanca Larraga-García, Aurora Pérez-Jiménez, Santiago Ros-Dopico, Javier Rubio-Bolívar, Manuel Quintana-Diaz, Álvaro Gutiérrez

**Affiliations:** 1Escuela Técnica Superior de Ingenieros de Telecomunicación, Universidad Politécnica de Madrid, Av. Complutense 30, 28040 Madrid, Spain; aurora96_5@hotmail.com (A.P.-J.); s.ros@alumnos.upm.es (S.R.-D.); a.gutierrez@upm.es (Á.G.); 2Instituto de Investigación IdiPAZ, Hospital La Paz, C/ Pedro Rico, 6, 28029 Madrid, Spain; javier84.rubio@gmail.com (J.R.-B.); mquintanadiaz@gmail.com (M.Q.-D.)

**Keywords:** hemorrhagic trauma, clinical simulation, training, patient care

## Abstract

One of the main preventable leading causes of death after a trauma injury is the hemorrhagic shock. Therefore, it is extremely important to learn how to control hemorrhages. In this paper, a hemorrhagic trauma simulator for lower limb has been developed and a pilot study has been accomplished to trail the simulator. Four different bleeding scenarios have been tested per participant, gathering information about the manual pressure exerted to control the bleeding. Data, altogether, from 54 hemorrhagic scenarios managed by final year medical students and doctors were gathered. Additionally, a post-simulation questionnaire, related to the usability of the simulator, was completed. All the participants managed to control the simulated bleeding scenarios, but the pressure exerted to control the four different scenarios is different depending if the trainee is a student or a doctor, especially in deep venous hemorrhages. This research has highlighted the different approach to bleeding control treatment between medical students and doctors. Moreover, this pilot study demonstrated the need to deliver a more effective trauma treatment teaching for hemorrhagic lesions and that hemorrhagic trauma simulators can be used to train and evaluate different scenarios.

## 1. Introduction

### 1.1. Traumatic Injury

The traumatic injury is one of the 10 leading causes of death and disability in the world, being the main cause of death for people under 35 years old [1,2,3]. Accidents, which are the main cause of traumatic injuries [2], are placed in the fifth position in the ranking of main causes of death in Europe [4].

Within the traumatic injuries, the hemorrhagic shock is the main preventable cause of death [5]. Therefore, detecting hemorrhages as soon as possible and learning different strategies to control them is key. Traumatic injuries occur initially in a prehospital environment and, depending on the country, the treatment of how to deal with a hemorrhagic scenario may vary [6,7,8]. In the military environment, protocols on how to deal with hemorrhages are more spread out. Therefore, some of the actions used in this military environment can be adopted in the civilian environment, especially after the events that promoted the Hartford consensus which highlights that everyone could save a life [9]. In this way, actions like the campaign “Stop the Bleed” [10,11,12,13] take momentum, approaching training to laypersons so they could learn how to identify different types of hemorrhages and how to control them.

### 1.2. Background

Massive hemorrhage may lead to hemodynamic instability, decreased tissue perfusion, organ damage, and death [14]. The main goals of resuscitation are two: restoring circulating blood volume and stopping the source of the hemorrhage [15]. In order to decrease the number of deaths, clinicians must be able to rapidly identify and manage hemorrhages. Recognizing the type of bleeding is needed to apply the appropriate hemorrhage control technique. Moreover, if the type of bleeding is recognized, the risks associated with it could be understood and the blood loss could be estimated [16]. Depending on the lesion suffered, three types of hemorrhages can be distinguished:**Arterial:** The arterial bleeding presents a bright red blood due to the fact that it carries hemoglobin, blood rich in oxygen. Moreover, the bleeding happens in a pulsating way showing the intermittent heart rate.**Venous:** The venous bleeding presents a dark red color as the blood has a low amount of oxygen. The venous flow blood is slower than the arterial one and it is homogeneous.**Capillary:** These vessels are smaller than venous or arteries. Therefore, the blood loss will happen slower than for venous or for arterial bleeding. It is the most common bleeding and the least dangerous.

The bleeding depends on its localization. When some of these vessels are bleeding, it may be appreciated outside of the body, known as external bleeding. However, in other situations, it may occur inside some cavity of the organism, known as internal bleeding [17]. Moreover, a classification of hemorrhages could be made taking into account the amount of blood lost. Therefore, it is possible to make a differentiation between the degrees of the shock according to the amount of blood loss as shown in Table 1. Additionally, the speed at which blood loss occurs worsens the situation [18].

When there are hemorrhages with high intensity, the human body reacts generating a pathophysiological response which is shown in different ways: pale and cold skin, sticky sweating, tachypnea, yawning, thirst, fast and soft pulse [14]. Due to the loss of blood pressure, dizziness and even disturbances of consciousness can occur. Therefore, it is necessary to learn how to control hemorrhages.

To accomplish this objective, clinical simulation is a tool that allows to teach and train different techniques in a practical manner [19,20], gathering real time information and allowing to objectively evaluate the performance of the simulation. Moreover, different studies [21,22] have proved that competences can be quickly achieved using clinical simulation in a more effective manner than just using the traditional way of teaching or even better than using actors for the simulations. Due to the progress in clinical simulation, some hospital areas which involve circulatory dysfunctions such as trauma and burn treatment units, demand bleeding trauma modules to support them to reproduce hemorrhagic real cases [23]. Usually, full patient simulators are which provide the ability of producing a bleeding automatically, but the cost of such simulators is quite high [24]. Nevertheless, other simulators offer the possibility to incorporate trauma modules allowing trainees to practice appropriate treatments for hemorrhagic situations [25]. These modules have some disadvantages such as the lack of automation and realism as shown in Figure 1a. Finally, another option to simulate hemorrhagic scenarios are limb simulators, see Figure 1b, which are considered specific and low technology simulators, but which can include automation and feedback [26].

### 1.3. Purpose

The objective of this paper is to develop a lower limb hemorrhagic trauma simulator to allow training on the identification of different types of bleeding and on the actions to take to control each of them. Additionally, the simulator will provide real time information about the actions performed, allowing to objectively evaluate each simulation. This represents an important improvement from the rest of the hemorrhagic simulators that are used in programs such as “Stop the Bleed” in which an important number of trainers is needed in order to visually evaluate how the different techniques are applied to control the hemorrhages [27,28]. An automated hemorrhagic trauma simulator would support an objective evaluation without the need of having a large number of trainers involved, which entails a reduction in the workforce and budget.

Therefore, to accomplish this objective, the lower limb simulator needs to be, first of all, designed and built. Then, it should be electronically managed and real time information needs to be generated with respect to the actions taken to stop the hemorrhage. This will allow the trainee to practice and gain ability to face hemorrhagic trauma scenarios. Moreover, this simulator will provide an objective evaluation.

The structure of the paper is as follows. The process followed to design and develop the hemorrhagic trauma simulator is explained in Section 2, together with the explanation of the pilot study performed to validate the simulator. Section 3 describes the results of the validation of the simulator in the pilot study. In Section 4, an analysis of the results obtained is presented. The paper is concluded in Section 5.

## 2. Materials and Methods

### 2.1. Hemorrhagic Trauma Simulator

A hemorrhagic trauma simulator has been designed and implemented for the lower limb.

#### 2.1.1. Components of the Simulator

Four lower limb main components have to be designed with the following characteristics:The bone: One of the main functions of the bones is to support the soft tissues, and the fulcrum of most skeletal muscles. Superficially, the bone is formed by a compact, smooth and very hard outer layer. Nevertheless, the bones internally are not completely solid, as they have many spaces. They have a porous structure formed by trabeculae. The femur is the main bone in the thigh, and it is the largest bone in the human body. It is the base of the simulator and therefore, it must be resistant enough to support the actions that will be performed on the simulator without deforming.The muscles: They are between 35% and 45% of the total weight of the body. The mechanical behavior of the muscles can be described through a basic model that involves elastic and contractile elements. In addition, they have a protective function of the bones. They are in charge of defining the shape of the simulator and to generate a firm structure.The blood vessels: The blood vessels are in charge of supporting the pressure exerted by the blood when it travels through them. They are in charge of conducting the blood in order to reproduce the different bleeding scenarios. There are many vessels in the thigh but only the main veins and arteries will be considered in this simulator: the deep and superficial femoral artery, the femoral vein and the greater saphenous vein. Moreover, they must be located in the precise place in the thigh to create a correct simulation.The skin: It is the largest organ of the human body with a dimension from 1.5 m^2^ to 2 m^2^ and a weight from 3 kg to 4 kg. Its structure is made of multiple layers, which makes its deformation behavior complex. The main functions of the skin are protection, reparation and adaptation. It has three superimposed layers: the epidermis, the dermis and the hypodermis. The epidermis is the most superficial and thin layer of the skin. It is joined to the following layer, the dermis, through a basement membrane to which it is firmly attached, and which provides the smooth appearance and texture of the skin. The dermis provides resistance and elasticity to the skin. It is the thickest layer of the skin and it is like a soft netting. Finally, the hypodermis layer is placed behind the dermis. All of them can be seen in Figure 2. Creating a skin as real as possible has been one of the main goals in the mechanical design. It is the superficial layer of the emulator and the real appearance of this layer is an important aspect as clinicians are in direct contact with it. For this reason, the touch should be very similar to the real skin.

#### 2.1.2. Materials Selection

The materials chosen to simulate each of the biological components are the following:Polymer: As previously mentioned, the bone is in charge of providing support as it is the main structure of the simulator. Therefore, the chosen material must have the functionality to be resistant enough. The material used is a biodegradable polymer derived from lactic acid, called polylactic acid or PLA, whose mechanical characteristics are appropriate to achieve the required functionality. On the one hand, this polymer has a glass transition temperature of 60 ºC. This temperature allows to have a wide range of working temperatures in which the material maintains its stiffness. Moreover, PLA is a biodegradable thermoplastic derived from renewable resources, which makes it a more environmentally friendly solution than other petrochemical-based plastics. In this way, bone made from PLA fulfills the required structural support function.Silicone: The muscles and the skin have been manufactured using platinum silicones: PlatSil Gel-25 and PlatSil Gel-0030. The main advantages of silicone-based models are related to the broad range of properties that can be simulated: easy manipulation, non-toxicity during and after preparation and long-term stability. Besides, models manufactured using silicones are durable and can be molded to obtain various shapes, from simple geometries to anatomical shapes.On the one hand, PlatSil Gel-25 has been chosen to simulate muscles because it is a silicone with appropriate hardness. Taking into account that the muscles in this simulator have the goal to support the skin, it is a good option. In addition, it cures quickly, only within 4 h, which allows to work with it quite fast.On the other hand, PlatSil Gel-0030 has been chosen to simulate the skin taking into account that its properties need to be similar to the human skin. Moreover, the study in [30] shows an experiment in which silicones with the same properties as PlatSil Gel-0030 offer similar mechanical characteristics as the skin. The skin has three different layers, and they have different densities. Slacker is a tactile modifier which allows modifying the densities of the silicones to be able to manufacture different layers as simulating the skin. For this reason, PlatSil Gel-0030 has been mixed with Slacker to reproduce the look, feel and touch of the living tissue.Plastic tubes: The blood vessels will be represented by intravenous lines used in hospitals. These are flexible plastic tubes with an internal diameter of 3.5 mm and an external diameter of 5 mm which allow fluids to be transported through them.

#### 2.1.3. Manufacturing Process

The manufacturing process followed to build and shape all the parts of the simulator will be presented hereafter:3D printing: The manufacturing of the bone with polylactic acid polymer is done by 3D printing. For this purpose, the Prusa i3 MK3S printer was used. This manufacturing system has provided the bone with the desired morphology.Molding: The soft parts made using silicones will be shaped using different molds. This technique will allow to create different complex morphologies. The use of molds to create silicone objects is one of the modern manufacturing techniques to construct parts that are used directly as finished products in which the post-processing is not necessary. The system is low-cost as specific machines are not required and moreover, the molds can be reused. The molds used have been made of paperboard but, in this process a wide range of materials could be used such as plastic, wood, or cork among others [31].

Taking into account the manufacturing process explained, the 3D printing of the femur has been done first. Then, the different silicones have been used to implement the muscles and the skin. For the skin, three different layers have been built to simulate the epidermis, the dermis and the hypodermis by mixing silicones with different densities thanks to the Slacker. In order to mix the components appropriately, the dimensions and the properties of the layers of the skin have to be considered. The most elastic and thick layer is the dermis. Taking this into account, the hypodermis and the epidermis do not need more elasticity than the one obtained using PlatSil Gel-0030. Therefore, all the amount of the Slacker has been used for the dermis. Each canister of silicone has two parts, A and B, which have to be mixed equally to get a perfect mixture. In Table 2, the different mixtures of Slacker and silicone will be presented showing different results. The desired effect for the dermis layer would be very tacky and therefore, the proportion used to mix the silicone with the Slacker has been the second option shown in Table 2. In this way, the touch of the skin obtained is really similar to the real touch of the skin.

Additionally, to simulate the different blood vessels that irrigate the lower limb, endovenous tracts have been incorporated in different areas of the lower limb taking into account the real location of the blood vessels. After the design of all these parts, they are implemented altogether in the prototype shown in Figure 3.

Then, two water-resistant hydraulic pumps are used to simulate the different hemorrhages depending on the hemorrhagic scenario to train. Each pump is immersed in either arterial or venous blood depending on the corresponding blood vessel to simulate. This blood is stored in two different tanks and it is simulated with dyeing water. Finally, pressure sensors [33] are included in the prototype to measure the actions taken in the simulator to control the hemorrhage. Then, all the components are electronically controlled by a microcontroller providing automation to the simulator. The microcontroller provides an output signal which activates the hydraulic pumps and then, the bleeding scenario to train starts. Then, once the trainee is managing the bleeding scenario, the microcontroller received an input signal coming from the force sensors. All the data obtained during the simulation are gathered, allowing an objective analysis of the simulation performed.

### 2.2. Hemorrhagic Scenarios

Four different hemorrhagic trauma scenarios have been designed in the lower limb to allow testing the simulator. The technique to train is the application of manual pressure on the simulated wound to stop the bleeding. The trauma scenarios are shown in Table 3 and they correspond with hemorrhages as a consequence of a lesion in the femoral vein, in the external saphenous vein, in the internal femoral artery and in the external femoral artery.

These blood vessels have been chosen as they represent hemorrhagic trauma scenarios that could happen when the lower limb is affected and that could compromise the life of the patient. The trainer selects the hemorrhagic scenario to train which could correspond with an internal or with an external blood vessel injury. This selection is always done as the pressure to stop the bleeding is not only dependent on the type of blood vessel, artery or vein, but also on the location of the vessel on the lower limb, internal or external. Therefore, the type of blood vessel and the location are the two main aspects that the trainer needs to set before the bleeding scenario starts. As the hemorrhages are different, the pressures to control them are also different.

This simulator allows to visually distinguish the scenarios as the arterial bleeding will be pulsatile and the venous one will be continuous; additionally, the color of the blood will be also different. Moreover, the simulator will allow to measure the actions performed so that trainees could practice the pressure to apply to stop the bleeding by knowing the real pressure they are applying along the simulation and the impact that it has on the hemorrhage, depending on the scenario performed. In case an evaluation on the technique applied is needed, this evaluation could be done in an objective manner, analyzing all the data stored during the simulation. Once the scenario to simulate is set, the bleeding will start, and the hemorrhage could be manually controlled.

### 2.3. The Pilot Study

The pilot study took place in the Hospital La Paz Institute for Health Research, IdiPAZ, gathering data from 32 simulations of final year medical students and 22 simulations from doctors. The project was introduced during trauma seminars for medical students, residents and doctors and the people that finally participated in the pilot study did it voluntarily after those seminars. The data was gathered during two weeks in November 2020 and this study was approved by the Ethics Committee of the Hospital Universitario La Paz (HULP: PI-3361).

The pilot study started with an introduction in which the simulator was shown to all the participants with a brief 15 min explanation before the first scenario starts. It is explained that the simulation scenarios present a lesion on the lower limb, that a hemorrhage will appear and that they would need to act accordingly. Reference to the trauma seminar is made to refresh some of the concepts on hemorrhagic trauma management provided. Then, the four different scenarios are accomplished sequentially providing a one-minute break between them. Once the four scenarios are finalized, the data about the pressure exerted to stop the bleeding was analyzed for both groups, medical students and doctors.

Then, a post-simulation questionnaire is provided to all the participants in order to gather feedback with respect to the usability of the simulator. This is done in a separate room so that participants can answer to the questionnaire while other participants are testing the simulator and were not influenced by the presence of the designers. The questionnaire consists of a set of closed questions concerning their attitude and perception of the learning experience on a 7-point Likert-type [34] scale ranging from strongly disagree to strongly agree. The questions addressed are shown in Table 4 and are the ones in accordance with the ethic committee directives. First of all, a set of questions was prepared by the engineers which was then discussed with the clinicians and the simulation staff of the IdiPAZ. Taking into account the expertise of all the parties involved, a final set of questions was defined.

## 3. Results

All the participants successfully completed all the scenarios presented with the exception of one of the participants that could only accomplish two of the four scenarios. This participant corresponds with one of the doctors that took part in this pilot study.

### 3.1. Hemorrhage Control Management

The data analyzed were the pressure exerted to stop the bleeding and if the hemorrhage was controlled or not. By doing this, the simulator could be validated taking into account both, the design and the implementation. With respect to the control of the hemorrhages, in all the simulated scenarios the hemorrhage was controlled. With respect to the pressure exerted, in the hemorrhagic trauma scenario in which the external saphenous vein is compromised, Scenario 2 according to Table 3, the medical students apply a much higher pressure than the doctors. The median of the pressure exerted by the students is of 275 mmHg whereas the median of the pressure applied by the doctors is of 156 mmHg. Figure 4a shows the wide range of values obtained depending on the doctor or on student that performed the pressure to stop the bleeding.

With respect to the femoral vein, Scenario 1 as presented in Table 3, it represents a hemorrhagic trauma scenario of an internal vein. The results obtained are shown in Figure 4b, in which a clear difference between the pressure applied by the students and by the doctors can be perceived. The range of pressures exerted by the doctors is smaller and the median is also much lower than the one exerted by the medical students to control this bleeding scenario. In this case, the difference is statistically significant, being the median pressure exerted by the students 468 mmHg and the one exerted by the doctors 131 mmHg.

With respect to the external femoral artery, Scenario 4 according to Table 3, in Figure 5a the pressure values exerted to stop these hemorrhages are shown. Also in this case, as in the previous ones, the doctors act more homogeneously, although in this case they apply a higher pressure than the medical students. In the previous scenarios, venous hemorrhages, the pressure exerted by the students has been always higher than the one exerted by the doctors. This may be due to the fact that doctors are more familiar with the fact that venous hemorrhages need a lower pressure to stop the bleeding than arterial hemorrhages. Therefore, the pressure applied by the doctors is higher as arterial hemorrhages need higher pressure values. To control the hemorrhage of the external femoral artery, the median of the pressure exerted by the doctors is of 286 mmHg whereas in the case of the medical students, the median value is of 215 mmHg.

In Figure 5b, the results obtained for a bleeding scenario of an injury of the internal femoral artery are shown, Scenario 3 as presented in Table 3. In this case, both groups apply a similar pressure. The median of the pressure exerted by the students is of 241 mmHg whereas the pressure exerted by the doctors is of 203 mmHg.

### 3.2. Usability Testing

Finally, the results obtained from the usability questionnaire are shown in Figure 6. In general, the response obtained is quite positive, being the median value of the answers between 6 and 7 (agree and strongly agree respectively) of the Likert scale, except for the questions 7, 9 and 11. In those questions, the median value of the responses provided by doctors are between 5 and 6 (somewhat agree and agree respectively).

## 4. Discussion

The goal of this pilot study was to demonstrate that the lower limb simulator can be used to train and evaluate the manual pressure technique used to face and treat different types of hemorrhagic trauma scenarios. Taking into account the results presented in the previous section, this goal is achieved and a clear need to train this technique is detected as the variation in the response in all the scenarios has been quite diverse, especially in medical students.

All the participants managed to control the hemorrhage; nevertheless, the results show that this is achieved using different approaches and therefore, this could be improved training on how to apply the needed pressure to stop different hemorrhages. This will make the treatment to the patient more efficient.

According to [35,36,37], the best technique to initially treat a hemorrhage is manual pressure. This pressure could be applied directly on the wound, without using any gauze or using a gauze or any other material that could absorb the blood. This is important as if a gauze is used, the pressure to apply to stop the bleeding will be higher than in the case that a gauze is not used. In this pilot study, the manual pressure has been exerted directly in the wound without using any gauze. Moreover, the best way to apply manual pressure is positioning one hand above the other and applying pressure with both hands [35]. In this pilot study, all the participants have controlled the hemorrhage using both hands and placing them as stated in [35]. If this technique is not sufficient to stop the bleeding, then it is recommended to use hemostatic agents and / or a tourniquet depending on the lesion; nevertheless, in the scenarios applied in the simulator, only the manual pressure technique is being validated.

Taking into account that the pressure to apply is different depending on the type of blood vessel and its location on the body [38,39], different pressures need to be applied to control hemorrhages. The arteries support a pressure of approximately 100 mmHg whereas the veins support a pressure of approximately 15 mmHg. This shows the difference in pressure that is needed to stop a hemorrhage depending on the type of blood vessel. Therefore, the simulator developed allows to distinguish the type of bleeding visually, providing realism to the scenario and facilitating the identification of the type of blood vessel affected that will lead to the decision on how much pressure to apply.

To stop a hemorrhage, it is needed to apply a higher pressure than the one the blood vessel is supporting. The pressure to apply to control an internal hemorrhage in the lower limb has to be around 100 mmHg higher than the pressure of the blood vessel damaged in the lower limb [40]. Therefore, the needed pressure to stop the hemorrhages could be seen in Table 1 and will be: 115 mmHg for the femoral vein, 35 mmHg for the external saphenous vein, 200 mmHg for the internal femoral artery, and 120 mmHg for the external femoral artery.

Therefore, the pressure to stop arterial hemorrhages needs to be higher than the pressure to stop venous hemorrhages [39]. Having a look at the results obtained, doctors apply a higher pressure to stop arterial hemorrhages whereas medical students apply a higher force in average to stop venous hemorrhages. Additionally, the applied pressure to stop internal hemorrhages has to be higher than the one needed to stop external hemorrhages [39]; in such cases, medical students apply a higher force for internal injuries whereas doctors apply similar pressure for both internal and external or even lower to stop an internal hemorrhage. This shows a clear need to train on how to better control hemorrhages. This is an important outcome, as there is not so much information available, as far as we know, on specific guidelines on pressure to exert depending on the blood vessel and the location of the vessels. For this reason, such information would complete the current protocols that exist with respect to hemorrhage control techniques.

Therefore, this study proposes an objective training and evaluation tool that will promote actions to train on hemorrhagic trauma management. This training could be not only focused on clinicians but also extended to paramedics, nurses and even laypersons. In the literature, there are publications in which laypersons are starting to be trained with a very positive outcome [9,10,11,12]. This is done using simulators which are not automated, and which do not allow objective evaluation of the techniques. Therefore, this simulator will have a positive impact on those trainings. Additionally, there are other publications that focuses on how to manage hemorrhagic lesions by clinicians [28,41,42]. In those papers, simulation is presented as a tool that supports learning on the management of hemorrhagic lesions. Nevertheless, none of them provide specific information on the pressure exerted during the simulation and the blood vessel affected. On top of that, this simulator is an easy tool to use and it could be adapted to the different levels of expertise proposing a tailored-made training and evaluation tool. Moreover, providing real time information to the trainees will improve their skills in a more effective manner as stated in [18,19]. This will improve the medical attention provided to hemorrhagic trauma patients as a quick response in hemorrhagic trauma scenarios is key.

Taking all this into account, some recommendations could be provided. First of all, further tests need to be done to fully support the results presented in the previous section. Then, the information about the pressures exerted to manage hemorrhages should be gathered and included in the hemorrhagic control guidelines by the clinicians. Moreover, more specific trainings should be prepared with those guidelines according to the experience of the trainees and the results obtained.

With respect to the usability questionnaire, the lowest score obtained corresponds to the questions number 7, 9 and 11. The question number 7 refers to whether the pressure needed to stop the hemorrhage is realistic or not. The feedback obtained by the doctors is that this could be improved specially incorporating the time needed to control a bleeding which is an important variable to consider when managing a venous or arterial bleeding. The question number 9 refers to whether the bleeding scenarios are realistic or not. The feedback obtained in this question was that the difference between the arterial bleeding (pulsatile) and the venous bleeding (continuous) was correctly perceived but that the difference in flow pressure should be higher and oozing out of the wound for venous bleeding. With respect to question 11, about the importance of showing a warning in case the pressure applied is too high. The punctuation provided suggests that this is perceived as not so important as the objective is to stop the bleeding and if the pressure is excessive, this could cause damages if it is maintained over time. Nevertheless, applying manual pressure on a lesion is not a technique that is maintained over time for 60–90 min which is the time above which it is consider that could be damaging for a patient [39].

## 5. Conclusions

This pilot study proved that the hemorrhagic trauma simulator developed is appropriate to train and evaluate the manual pressure technique used to control hemorrhages. A lower limb bleeding trauma simulator has been designed and developed, as well as an electronic circuit to check its functionality. The simulator developed is a mechanical device manufactured using different synthetic materials which have provided the adequate and desired features to look like a human lower limb. The simulator is able to set up the type of bleeding and, according to it, different pressures can be applied to manage the hemorrhage. In addition, the simulator does not only execute a bleeding trauma injury scenario, but it also provides feedback when the hemorrhage is controlled. Besides, the final appearance of the simulator has achieved a great similarity to reality, allowing a complete immersion in the simulated scenario. Moreover, the results obtained from the experiments confirm that further training may be required during medical school to train future clinicians on this technique.

### Future Directions

Further investigations need to be done to fully support this pilot study by testing this hemorrhagic trauma simulator with more bleeding scenarios and incorporating more sensors, allowing to train more techniques such as the placement of a tourniquet. Therefore, other sensors should be integrated which will collect data and presenting it in a form of dashboard for data analysis and statistics. Moreover, a web interface is under development in order to show, in a friendly manner, all the real time data produced during the simulation and gathered by the different sensors. In the future, it will be interesting to merge this simulator with augmented reality to offer a more immersive experience.

## Figures and Tables

**Figure 1 sensors-21-03816-f001:**
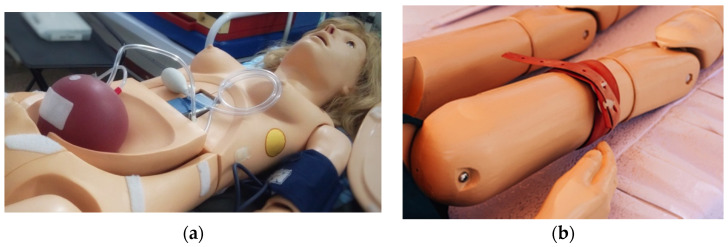
Trauma hemorrhagic simulators. (**a**) Individual bleeding trauma module [25]; (**b**) Lower limb trauma simulator [26].

**Figure 2 sensors-21-03816-f002:**
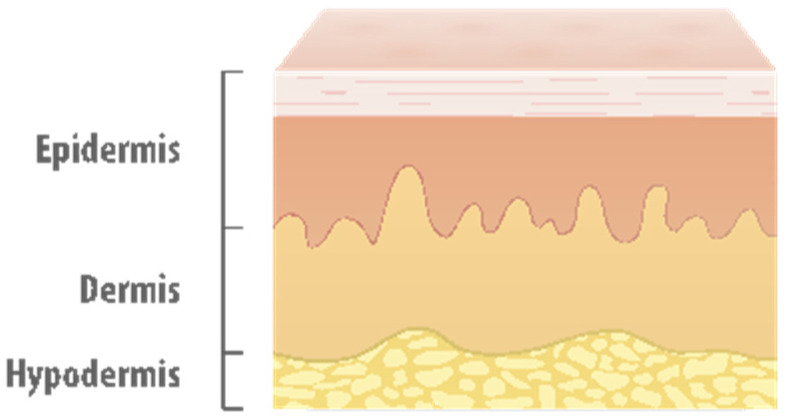
Layers of the skin [29].

**Figure 3 sensors-21-03816-f003:**
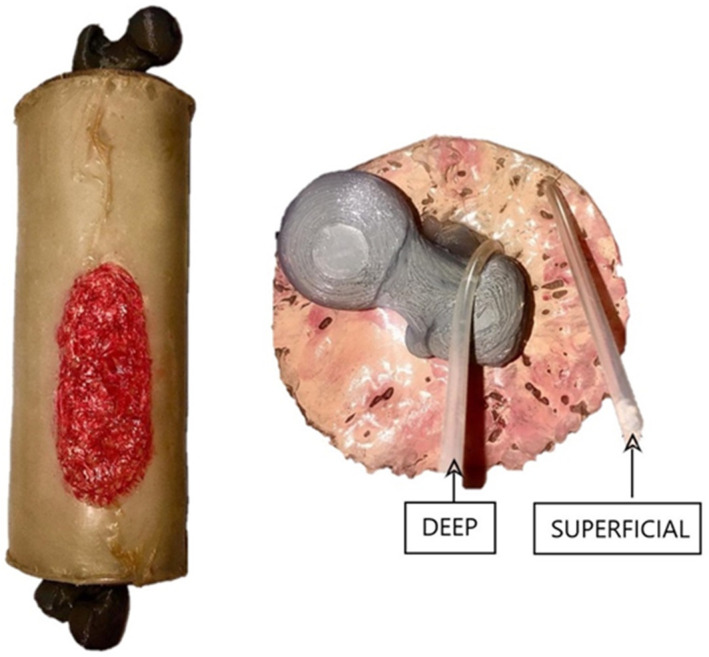
Prototype of the lower limb design and implementation in which the femur, the muscles and the skin are shown. In the left image, the simulator is shown with a lesion in the inferior part of the leg and in the right image, an axial section of the simulator is presented in which the location of different blood vessels is shown.

**Figure 4 sensors-21-03816-f004:**
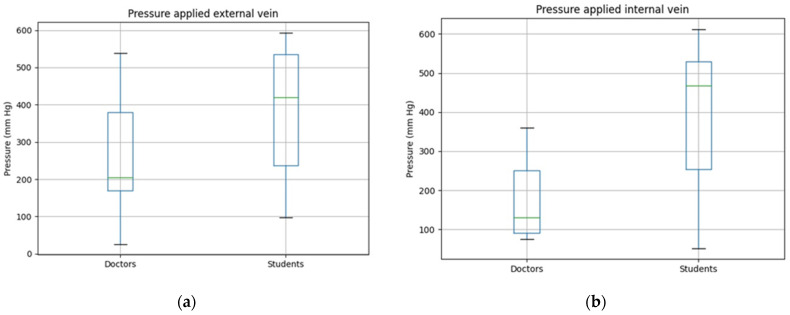
Pressure exerted to stop the bleeding by medical students and doctors if a venous bleeding occurs in the lower limb. (**a**) Pressure exerted to stop an external venous hemorrhage; (**b**) Pressure exerted to stop an internal venous hemorrhage.

**Figure 5 sensors-21-03816-f005:**
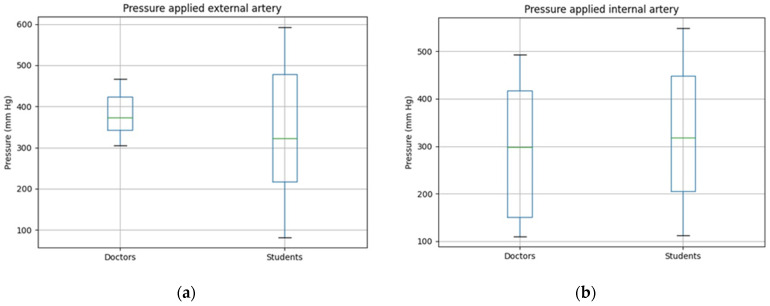
Pressure exerted to stop the bleeding by medical students and doctors if an arterial bleeding occurs in the lower limb. (**a**) Pressure exerted to stop an external arterial hemorrhage; (**b**) Pressure exerted to stop an internal arterial hemorrhage.

**Figure 6 sensors-21-03816-f006:**
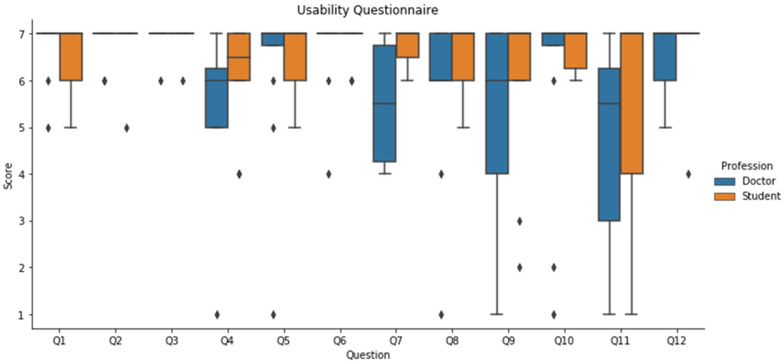
Usability post-simulation questionnaire. The values shown are scores provided by all the participants following the 7-points Likert scale. The responses provided by doctors are provided in blue and the ones provided by medical students, are shown in orange.

**Table 1 sensors-21-03816-t001:** Hemorrhagic shock classification [15].

	Class I	Class II	Class III	Class IV
Blood loss (mL)	<750	750–1500	1500–2000	>2000
Blood loss (%)	15%	15–30%	30–40%	40%
Pulse rate (beats/min)	<100	>100	>120	>140
Blood pressure	Normal	Decreased	Decreased	Decreased
Respiration rate (breaths/min)	14–20	20–30	30–35	35–40
Urine output (mL/h)	>30	20–30	5–15	Negligible
Mental status	Normal	Anxious	Confused	Lethargic

**Table 2 sensors-21-03816-t002:** Different mixtures of Slacker with silicones in order to obtain different results. As each part corresponds to an equal amount of product [32].

Mixture	Results
33.3% Part A silicone + 33.3% Part B silicone+ 33.3% Slacker	Tacky
25% Part A silicone + 25% Part B silicone+ 50% Slacker	Very Tacky
20% Part A silicone + 20% Part B silicone+ 60% Slacker	Extremely Tacky
16.6% Part A silicone + 16.6% Part B silicone+ 66.6% Slacker	Super Soft Tacky Silicone Gel

**Table 3 sensors-21-03816-t003:** Scenarios in which the lesion could happen classifying the blood vessel depending on the type, vein or artery and on the location of the blood vessel, internal or external. Additionally, the needed pressure to stop the bleeding is stated in mmHg.

Scenario	Blood Vessel	Location of Blood Vessel	Pressure Exerted to Stop Bleeding (mmHg)
1	Femoral vein	Internal	115
2	External saphenous vein	External	35
3	Internal femoral artery	Internal	200
4	External femoral artery	External	120

**Table 4 sensors-21-03816-t004:** Questions of the usability questionnaire.

Questions
Q1. Do you consider that this tool is useful to train protocols to treat hemorrhages?
Q2. Do you consider that this tool is useful as a training tool?
Q3. Do you think that the simulator motivates learning?
Q4. How would you score your experience with the simulator?
Q5. Do you think that this tool support critical thinking and decision making? Q6. Do you consider important that the blood flow would be proportional to the pressure exerted? Q7. Is the pressure needed to stop the hemorrhage realistic? Q8. Do you consider that the main blood vessels of the lower limb have been simulated? Q9. Are the different hemorrhagic scenarios realistic? Q10. Do you think that it would be important to monitor the time during which the pressure is maintained on the wound? Q11. Do you think that it is important to show a warning when the applied pressure is too high? Q12. Overall, I am satisfied with the tool.

## Data Availability

Data sharing not applicable.

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
