# Peer review of "Design and Development of a Hemorrhagic Trauma Simulator for Lower Limbs: A Pilot Study"

_sensors, 2021, doi:10.3390/s21113816_

Round 1
Reviewer 1 Report
Suggests improving the objectives of the paper
Suggests improving the conclusions
Missing DOI for publications
the publication should include scenarios
Reviewer 2 Report
NOTES TO THE AUTHOR(S)
This is an important study in an under-researched area of the world. However, it needs a considerable amount of work to be publishable. Some areas need clarification as noted below:
Introduction
- The paper does not cite an appropriate range of literature sources. Some statements need references. Further, there is no clear distinction between manuscript sections in terms of the content they report. First, I suggest dividing the section "Introduction" into three components, respectively introduction (explain the general argument of the paper, without going into specific details) background (situate the study concepts within the context of extant sustainability knowledge, discuss the international relevance of the concepts) and purpose, creating greater clarity in the analysis of the reader. The authors should revisit the text and, if wishing to retain the structure of the argument, at least explain clearly in the introduction how the argument proceeds and which are the steps they have taken in answering the research question.
Information on the pilot study is very scarce, namely:
a) The ethical aspects in collecting data are not specifically clarified. Variables such as the voluntary nature of the subjects´ participation, the approval by the local IR, the offer of incentives to participate (how participants were compensated for participation), sharing and use of data are not patent. More information is needed about the issues around informed consent or confidentiality or how they have handled the effects of the study on the participants during and after the study.
b) When the data were collected? Data collection period? How the research was explained to respondents/participants?
c) More information is needed about the clinical simulation context. Were clinical vignettes used? It would be interesting to present an example.
d) Usability questionnaire: How were the questions chosen? What was the process? The interview guide was developed based on instruments previously used in other studies?
Discussion
- The discussion appears to relay the main findings, and there is some discussion of what the findings mean, but there was very little attention paid to how this study might relate to our existing knowledge base, and little attention paid to how the current findings might extend our current knowledge.
- How valuable is the research? Do they consider the findings in relation to current practice or policy, or relevant research based literature.
- In the discussion section, identify recommendations for practice/research/education as appropriate, and consistent with limitations.
Organization and style:
- The manuscript will serve a broad audience of students, researchers, and practitioners, however, the manuscript needs to be carefully and attentively proofread, because some sentences are awkwardly constructed, punctuation is deficient, and therefore reading is occasionally difficult to follow. Would recommend a thorough technical edit of this paper.
Reviewer 3 Report
-The paper presents a hemorrhagic trauma simulator for lower limb and a pilot study from the trial of the simulator.
-The authors reported results that seem to demonstrate the need to these type of simulations in the context of hemorrhagic trauma treatment for medical students.
-It was presented a good background about hemorrhagic shock that could be useful for readers not familiar with the topic or area.
-Have the authors considered the use of augmented reality in addition to the simulator presented? That could give an even more immersive experience useful in particular to medical students.
-I think the authors could also integrate other type of sensors in the simulator that collect data and presents it in a form of a dashboard for data analysis and statistics.
-Line 242: I think it should be "the" instead of "de" at the start of the line. 
Round 2
Reviewer 2 Report
The authors have faithfully revised the paper according to the content of the review. I would like to agree to the publishing as it is.